# Fencing vs. Grazing: Divergent Effects on Soil Seed Bank Structure and Grassland Recovery Pathways in Northern Tibetan Alpine Grasslands

**DOI:** 10.3390/plants14060900

**Published:** 2025-03-13

**Authors:** Yuyuan Xie, Yongjie Liu, Wencheng Li, Ningning Zhao, Xuehao Li, Yifan Chen, Guozhi Lai, Xin Lou, Xiangtao Wang, Xuehong Wei

**Affiliations:** 1College of Animal Science, Xizang Agricultural and Animal Husbandry University, Nyingchi 860000, China; 2Southwest Survey and Planning Institute of National Forestry and Grassland Administration, Kunming 650216, China; 3Qiangtang Alpine Grassland Ecosystem Research Station (Jointly Built with Lanzhou University), Xizang Agricultural and Animal Husbandry University, Nyingchi 860000, China; 4Key Laboratory of Biodiversity and Environment on the Qinghai-Xizang Plateau, Ministry of Education, School of Ecology and Enviroment, Xizang University, Lhasa 850000, China; 5State Key Laboratory of Herbage Improvement and Grassland Agro-Ecosystems, Center for Grassland Microbiome, College of Pastoral, Agriculture Science and Technology, Lanzhou University, Lanzhou 730000, China; 6Yunnan Salt Industry Dianzhong Company, Kunming 650000, China; 7School of Life Sciences, Guizhou Normal University, Guiyang 550025, China

**Keywords:** soil seed bank, grazing exclusion, graze, alpine grassland, northern Tibet

## Abstract

Alpine grasslands are a critical component of the Qinghai–Tibet Plateau ecosystem, but their soil seed bank (SSB) patterns and driving mechanisms remain unclear under the influence of climate change and human activities. This study analyzed grazing exclusion (via fencing) and grazing effects using 12 sites in the alpine steppe (AS) and alpine desert steppe (AD) in northern Tibet to analyze the effects of fencing and grazing management, as well as hydrothermal and soil factors, on the SSB density and diversity. Linear regression models were applied to explore the relationships between the SSB density and environmental factors, while comparisons of the management modes revealed the potential impacts of fencing. The results show that fencing significantly increased the SSB density and diversity, especially in the AS, while grazing negatively impacted the SSB density and the Pielou evenness index. Hydrothermal factors strongly influenced the SSB in the AS, with the density positively correlated with precipitation and negatively with temperature, while responses in the AD were weak. Soil factors, such as the available phosphorus (SAP) and available potassium (SAK), were key to SSB formation in the AD, whereas ammonium nitrogen (NH4_N) and the pH were critical in the AS. Fencing optimized the hydrothermal conditions and nutrient availability, promoting SSB recovery, though its effects varied between the grassland types. This study provides scientific insights for alpine grassland restoration and sustainable management.

## 1. Introduction

The Qinghai–Tibet Plateau is one of the most sensitive and fragile terrestrial ecosystems in the world [1,2]. Alpine grasslands, which cover more than 50% of the plateau, represent the largest vegetation type in this region [3]. However, under the dual pressures of climate change and human activities, alpine grasslands are undergoing widespread degradation, resulting in a series of issues, such as vegetation loss, soil deterioration, and the decline of ecological functions [4,5]. Fencing, as a low-disturbance and sustainable grassland restoration approach, has been widely applied for the recovery of alpine grasslands due to its ability to enhance ecosystem stability, biodiversity, and resistance to disturbances [6,7]. Meanwhile, the soil seed bank (SSB), serving as the primary source of sexual reproduction in grassland ecosystems, plays a critical role in the formation of potential plant communities, adaptation to extreme environments, and buffering against ecological disturbances [8,9]. Especially in the low-temperature environment of the Qinghai–Tibet Plateau, where permafrost covers approximately 70% of the region, the cold conditions prolong seed longevity and reduce the seed decomposition rates [10]. This makes the SSB a crucial entry point for understanding and restoring the dynamics of alpine grassland ecosystems.

Although numerous studies have focused on the effects of fencing and grazing on SSBs, including seed density, diversity, and their relationship with the aboveground vegetation [11,12,13,14,15,16,17], most of these studies are limited to small-scale plots or specific grassland types; systematic research across large-scale regions and various grassland types is lacking. Additionally, the relationships between environmental factors (e.g., climate and soil nutrients) and SSB dynamics remains insufficiently explored. For example, questions, such as how hydrothermal conditions drive the formation of SSBs, the mechanisms by which soil nutrients (e.g., nitrogen, phosphorus, and potassium) influence SSBs, and how these environmental factors jointly regulate the density and diversity of SSBs under fencing and grazing management regimes remain unanswered systematically. Therefore, exploring the complex relationships between SSBs and environmental factors is crucial for addressing this research gap.

This study investigates 12 experimental sites of alpine grassland plots along a northern Tibetan Plateau transect, comparing the SSB dynamics (density and diversity) and their interactions with the aboveground vegetation under fencing vs. grazing, while evaluating the climatic and soil drivers. This study aims to address the following questions: (1) Does fencing influence the successional trajectory of alpine grassland ecosystems by regulating the SSB? (2) How does fencing management mediate the interactions between environmental factors and the SSB? By resolving how fencing mediates SSB–environment interactions in alpine systems, this work advances targeted restoration strategies for degraded Tibetan grasslands, advancing our mechanistic understanding of fencing-mediated SSB recovery in alpine systems.

## 2. Materials and Methods

### 2.1. Study Site

The northern Tibet region, located in the core area of the northwestern Qinghai–Tibet Plateau, covers a total area of approximately 446,000 square kilometers, accounting for 37.10% of the total area of the Tibet Autonomous Region. Also known as Qiangtang, it lies in the northern part of China’s Tibet Autonomous Region. It spans about 760 km from north to south (29°53′–36°32′ N), and approximately 1200 km from east to west (78°41′–92°16′ E). The northern Tibet region is characterized by zonal vegetation landscapes, such as an alpine steppe (AS), and serves as the core area of Tibet’s pastoral industry. Due to the plateau’s cold, arid, and extreme environmental conditions, the northern Tibetan Plateau is highly susceptible to the impacts of climate change and human activities, leading to ecological problems, such as grassland desertification, black-soil patches, and sparse vegetation [18,19,20]. From 2003 to 2012, a cumulative area of 86,000 km^2^ of grazing exclusion fencing was established in the northern Tibetan Plateau, accounting for approximately 10% of Tibet’s usable grassland area [21]. As shown in Figure 1, this study is primarily based on the horizontal transect of alpine grasslands established by the Lhasa Plateau Ecological Research Station of the Institute of Geographic Sciences and Natural Resources Research, Chinese Academy of Sciences (2009–2019). This research focuses on the SSB in the region’s major AS and alpine desert steppe (AD). AS is dominated by *Stipa purpurea*, *Stipa capillacea*, and *Stipa subsessiliflora* var. *basiplumosa*, occupying regions with mean annual precipitation (MAP) of 250–450 mm and vegetation cover of 45–60%; AD is characterized by *Stipa purpurea*, *Ceratoides latens*, and *Stipa glareosa*, which have adapted to arid conditions (MAP < 250 mm) with sparse vegetation cover (15–30%) [22]. This transect is representative of a typical plateau continental climate, characterized by the Nagqu–Guoluo semi-humid climate zone, the Qiangtang semi-arid climate zone, and the Ali arid climate zone. The plant growing season spans from mid-to-late May to mid-to-late September, with the peak growing period occurring from June to August, during which 65–85% of the annual precipitation is concentrated. The region experiences significant interannual and diurnal temperature fluctuations. The mean temperature of the hottest month ranges from 6 to 10 °C, while the mean temperature of the coldest month is below −10 °C. The annual average temperature is generally below 0 °C, with daily average temperatures reaching above 5 °C from July to September, but dropping to freezing at night [23]. The basic conditions of the surveyed sampling sites are shown in Table 1. Notably, vegetation types for some sampling plots were determined by field validation of dominant species and local conditions, despite their geographic locations within mapped zones of different grassland types.

### 2.2. Experimental Design

At the 12 surveyed sampling sites, paired 100 × 100 m plots were established for free grazing and grazing exclusion (fence), with a distance of no more than 2 km between paired plots. Along the diagonal of each plot, five 10 × 10 m subplots were set up at intervals of approximately 20 m. Within each plot, the SSB, vegetation, soil, and other factors were surveyed and sampled. The surveys were conducted during early spring and late summer of 2018. Vegetation surveys and soil sampling were carried out in August, when plant community species diversity was at its peak, while the SSB survey was conducted in April of the same year, prior to the start of the growing season.

### 2.3. Aboveground Vegetation Investigation

Within each subplot, 3 randomly distributed quadrats were established for vegetation surveys, resulting in 360 total surveys (12 sites × [2 treatments × 5 subplots] × 3 quadrats). In the eastern meadow sites of northern Tibet, the sampling area was 0.5 × 0.5 m, while in the central steppe and western desert regions the sampling area was 1 × 1 m. Total vegetation cover (%) was estimated through multi-observer visual assessments using reference images (0–100% cover standards), with final values determined by majority agreement among three trained observers. Aboveground vegetation was clipped at ground level, sorted by species, and placed into envelopes for transport back to the laboratory, where they were air-dried to a constant weight. Subsequently, the total aboveground biomass (AGB) and biomass of individual species were measured.

Further, the importance value (IV, Equation (1)) is an important quantitative indicator of a species. In plant communities, relative cover ($C_{r}$) and relative biomass ($B_{r}$) are used to calculate the IV of species [24], and the formula is as follows:(1)$$IV=(C_{r}+B_{r})/2$$
where $C_{r}$ is relative cover ($C_{r}$ = cover of a species in the plant community/cover of all species × 100%) and $B_{r}$ is relative biomass ($B_{r}$ = biomass of a species in the plant community/biomass of all species × 100%).

Species richness (S, i.e., species count), Shannon–Wiener index (H, Equation (2)), Simpson index (D, Equation (3)), and Pielou evenness index (E, Equation (4)) are used to characterize the species diversity of plant communities [25], and the formulas are as follows [26]:(2)$$H=-\sum_{i=1}^{S}\left(P_{i}\ln P_{i}\right)$$(3)$$D=1-\sum_{i=1}^{S}P_{i}^{2}$$(4)E=H/lnS
where S is the number of species visible in a plot or community; $P_{i}$ is the relative importance of species i (Equation (5)), which is calculated as follows:(5)$$P_{i}=IV_{i}/\sum IV$$

### 2.4. Soil Seed Bank Sampling

SSB samples were collected within each subplot using an S-shaped sampling method. Mixed soil samples were obtained randomly using a soil auger (inner diameter of 3.5 cm); six soil cores (0–20 cm depth) were homogenized to form one composite sample per quadrat, resulting in 120 analytical units (12 sites × [2 treatments × 5 subplots] × 1 composite quadrat). Litter and roots were removed, and the samples were placed into labeled, sealed plastic bags for storage in a cool environment. The samples were then transported to the laboratory for subsequent germination identification and counting of species in the SSB.

We quantified the SSB species composition through germination assays. Seedlings were identified using the *Flora of Tibet* [27] and cross-validated against field-collected seed specimens. Identified seedlings were removed upon recording; unidentifiable individuals were retained until no new germination occurred for 14 consecutive days.

The calculation methods for SSB indicators are consistent with those for plant community characteristics. In an SSB, the relative importance value (IV) of a species is defined as the proportion of seeds of that species to the total seed count. The calculation of species diversity-related indices is detailed in Section 2.3. Additionally, the seed bank density mentioned in the text has been converted to the total number of seeds per unit area (seeds/m^2^).

### 2.5. Measurement of Environmental Factors

The collection of soil samples followed the method described in Section 2.4, where six cores were combined into a composite sample and air-dried for subsequent analysis. Specifically, soil organic matter (SOM, %) was measured using the dichromate method, total nitrogen (STN, g/kg) was determined by the Kjeldahl method, and total phosphorus (STP, g/kg) was analyzed using the sodium hydroxide fusion–molybdenum–antimony colorimetric method. Ammonium nitrogen (NH4_N, mg/kg) and nitrate nitrogen (NO3_N, mg/kg) were extracted using a 2 mol·L^−1^ KCl solution at a 1:5 ratio and measured with a continuous flow analyzer. Soil available phosphorus (SAP, mg/kg) was measured using the NaHCO_3_ extraction–molybdenum–antimony colorimetric method, while available potassium (SAK, mg/kg) was determined using the ammonium acetate extraction–atomic absorption method. Soil pH was measured using a pH electrode at a soil-to-distilled water ratio of 2.5:1.

For 2018 monthly climatic records (January–December), we obtained climate information for the northern Tibet sampling area through the China Meteorological Science Data Sharing Service Network (http://cdc.cma.gov.cn/home.do, accessed on 11 March 2019) and the HOBO microclimate automatic observation stations in the northern Tibetan alpine grassland. We downloaded the growing season (May to September) temperature and precipitation data relevant to the sampling sites from this network. Using the professional climate data spatial interpolation software ANUSPLIN 4.3 in ArcView 3.2, we performed data interpolation and validated the interpolated raster data using measurements from the ten HOBO microclimate stations in the area. From ArcGIS 9.2, we extracted the annual precipitation (APre, mm), mean annual temperature (MAT, °C), growing season precipitation (GSPre, mm), and growing season mean temperature (GSMT, °C) for the sampling sites based on latitude and longitude information obtained from GPS.

### 2.6. Statistical Analysis

First, Excel was used for the basic processing of the experimental observed data. The Shapiro–Wilk test was then performed in R (Version 4.4.1; R Development Core Team, Vienna, Austria) to assess the normality of the data. Based on this, we utilized the “*psych*” (v2.4.12) package in R to conduct Spearman correlation analysis, exploring the relationships between SSB density and various factors. We applied Wilcoxon rank-sum tests to determine the differences in indicator variables under enclosure and grazing patterns. These included Seed_density, (SSB density), APre, MAT, GSPre, GSMT, SOM, STN, STP, SAP, SAK, NO3_N, NH4_N, S_seed (species richness of SSB), H_seed (Shannon–Wiener index of SSB), D_seed (Simpson index of SSB), E_seed (Pielou evenness index of SSB), Cover, AGB (aboveground biomass), S_pl (species richness of aboveground vegetation), H_pl (Shannon–Wiener index of aboveground vegetation), D_pl (Simpson index of aboveground vegetation), and E_pl (Pielou evenness index of aboveground vegetation).

We primarily used R for data analysis and visualization (“*ggplot2*” packages, v3.5.1) presented in this paper. The splitting of the violin plot was achieved using the “*introdataviz*” (v0.0.0.9003) package, while PowerPoint was utilized for formatting and enhancing some images.

## 3. Results

### 3.1. Spatial Patterns of Soil Seed Banks in Two Grassland Types Under Different Management Modes

As shown in Figure 2, under fenced management, the seed densities in both the AD and AS were significantly higher than under grazing management. Significant differences in the seed density between the fenced and grazed plots were observed in both the AD and AS (AD: *p* < 0.001; AS: *p* < 0.01), indicating stronger management sensitivity in the AD. Despite this, the AS maintained higher seed densities than the AD across all treatments, demonstrating that fencing consistently promotes seed bank enrichment regardless of the ecosystem type.

Among the soil factors, the SAP showed significant differences in the AD (*p* < 0.01), with higher values under fenced management compared to grazing management. This suggests that fenced management may promote phosphorus accumulation or reduce its loss. No significant differences were observed for the other factors (such as the SOM, STN, STP, SAK, pH, NO3_N, and NH4_N) between the two management modes.

Under fenced management, the vegetation cover and aboveground biomass (AGB) in the AD were significantly higher than under grazing management, indicating that an alpine desert steppe (AD) responds more sensitively to fenced management. For the AGB indicator, fenced management of the AD resulted in a significantly higher value than grazing, but there was no significant difference in the AGB response to fenced management in the AS. This may be due to the already high productivity baseline of the AS, which did not show significant additional growth under the management mode’s regulation.

The seed diversity indicators (S_seed, H_seed, D_seed, and E_seed) did not show significant responses to the different management modes, indicating that the management mode has a weak regulatory effect on seed bank diversity, richness, and seed distribution. Similarly, the vegetation diversity indicators (S_pl, H_pl, and D_pl) also did not exhibit significant differences between the two management modes.

As shown in Figure 3, the experimental results indicate significant differences in the influence of hydrothermal factors, soil factors, plant-related factors, and seed-related factors on the SSB between the AS and AD grassland types. In the AS system, precipitation (APre and GSPre) and plant-related factors significantly promote the SSB, while temperature (MAT and GSMT) and certain soil nutrients (STP, SAK, and pH) significantly inhibit the SSB. The vegetation and seed diversity indicators generally promote the stability of the SSB, highlighting their importance for ecosystem structure. In the AD system, the impact of the hydrothermal factors is weaker, and the SSB primarily depends on soil nutrients (e.g., STP, SAP, and SAK). The effects of fenced and grazing management on certain factors (e.g., SAP, SAK, NO3_N, E_seed, and AGB) show differences in their directions and significance, reflecting the different regulatory mechanisms of the management modes and grassland types on the SSB.

Specifically, the seed density in the AS system shows a significant positive correlation with the precipitation indicators (APre and GSPre; *p* < 0.001), and under fenced management, precipitation significantly increases the seed density in the AS system. This indicates that precipitation is one of the key factors influencing the formation of the SSB in the AS. In contrast, the seed density in the AD system does not respond significantly to precipitation (*p* > 0.05), suggesting a lower dependence of the AD SSB on precipitation. The seed density in the AS is significantly negatively correlated with the temperature indicators (MAT and GSMT; *p* < 0.001), indicating that high temperatures may suppress the formation of the SSB; conversely, the seed density in the AD is positively correlated with temperature, but the correlation does not reach significance (*p* > 0.05; Figure 3). Compared to the AD, the SSB in the AS is more sensitive to hydrothermal factors, while the AD SSB is less directly regulated by hydrothermal factors and shows a lower dependence on them. Its ecological adaptability may rely more on other environmental factors, or the inherent characteristics of the grassland type itself.

In the AS system, the effects of the soil nutrient indicators on the SSB show clear pattern differences in their direction and significance. Both the STP and SAK are significantly negatively correlated with the seed density under both management modes in the AS system (*p* < 0.001), suggesting that higher levels of STP and SAK may inhibit the formation of the SSB. This inhibitory effect may be related to plant competition triggered by excessive soil nutrients. The SAP is significantly positively correlated with the seed density in the AD system (*p* < 0.01), but significantly negatively correlated with it in the AS system (*p* < 0.05), which may reflect differences in soil nutrient utilization efficiency between the grassland types. Additionally, NO3_N is not significantly correlated with the SSB under fenced management in the AS system (*p* > 0.05), but is significantly negatively correlated with it under grazing conditions (*p* < 0.05), indicating that a grazing disturbance may cause nitrogen loss, inhibiting the formation of the SSB. NH4_N is significantly positively correlated with the SSB in the AS system (*p* < 0.01), suggesting that moderate levels of NH4_N facilitate the formation of the SSB. In contrast, NO3_N and NH4_N show no significant correlation with the seed density in the AD system (*p* > 0.05), likely because the SSB of the alpine desert steppe is more adapted to or less dependent on nitrogen. Overall, the AS system is more sensitive to changes in soil nutrients, while the AD system relies more on the input of soil phosphorus to support the formation of the SSB, with weaker influences from potassium and nitrogen.

The vegetation-related indicators, such as the AGB, are significantly positively correlated with the seed density under fenced management in the AS system (*p* < 0.01), suggesting that higher vegetation productivity contributes to increased SSB density. Additionally, the S_pl and H_pl significantly promote seed density in the AS system, while the vegetation-related factors (Cover, AGB, S_pl, H_pl, D_pl, and E_pl) in the AD system show no significant correlations overall (*p* > 0.05). This may be because the vegetation in the alpine desert steppe is sparse, and the SSB is less dependent on vegetation cover.

The seed-related indicators (E_seed, S_seed, and D_seed) show significant differences under the different management modes. In the AS system, the S_seed, H_seed, and D_seed are significantly positively correlated with the seed density under fenced management (*p* < 0.01), suggesting that seed richness and diversity are the key factors for enhancing SSB density. However, under grazing conditions, the E_seed is significantly negatively correlated with the SSB in the AS system (*p* < 0.05), indicating that the disturbances caused by grazing may disrupt the uniformity of the SSB. In the AD system, the effect of the seed diversity indicators on the SSB is not significant under either management mode (*p* > 0.05), reflecting that seed diversity may not be a key factor for SSB formation in the alpine desert steppe environment. Overall, the SSB in the AS system is more sensitive to the seed-related indicators, while the SSB in the AD system shows weaker responses to these factors and higher stability.

### 3.2. Response of the Soil Seed Bank to Hydrothermal Environmental Factors

The experimental results indicate significant differences in the response of the seed density to the APre between the AS and AD grassland types. In the AS system, the seed density is significantly positively correlated with the APre (*p* < 0.001, R^2^ = 0.741); especially when the APre exceeds 500 mm, the seed density significantly increases. This suggests that precipitation is a key driving factor for seed bank formation in the AS system. In contrast, the correlation between the seed density and the APre in the AD system is not significant (*p* = 0.515, R^2^ = 0.007), indicating that precipitation has a relatively small impact on the seed bank in the AD system (Figure 4). Additionally, the bubble size (MAT) shows that when the temperature increases in the AS system, the seed density significantly decreases, suggesting that rising temperature may inhibit seed bank formation. On the other hand, the seed density in the AD system shows little variation across temperature gradients, reflecting the weaker response of the seed bank to hydrothermal factors.

### 3.3. Response of the Soil Seed Bank to Soil Factors

The linear regression results (Figure 5) indicate differences in the responses of the seed density to the eight soil factors between the AD and AS systems. In the AD system, the SAP and STP under fenced management displayed a significant positive correlation, while the STP and SAK under grazing management showed a significant positive correlation (Table 2). In contrast, in the AS system, the STP, SAK, pH, and NO3_N showed a significant negative correlation with the seed density, while NH4_N exhibited a significant positive correlation (Table 2). The relationships between the remaining factors and the seed density were relatively weak, consistent with the results of the correlation analysis (Figure 3).

### 3.4. Response of the Soil Seed Bank to Seed Bank Attributes and Vegetation Factors

As shown in Figure 5, in the AD system, the overall relationships between the seed density and these indicators were weaker, while in the AS system, the seed density exhibited notable differences in response to the seed bank attributes (S_seed, H_seed, D_seed, and E_seed) and vegetation factors (Cover, AGB, S_pl, H_pl, D_pl, and E_pl). In the AD system, the relationships between the seed density and seed bank or vegetation factors were generally weaker, indicating a lower level of response. These results suggest that the dependence of the desert steppe seed bank on the seed bank attributes and vegetation factors is relatively low (Table 2). In contrast, in the AS system, the seed density showed significant positive correlations with the S_seed, H_seed, and D_seed, whereas, under grazing conditions, it exhibited a significant negative correlation with the E_seed. Among the vegetation factors, the S_pl, H_pl, and AGB demonstrated more pronounced positive responses to the seed density under the fenced management mode.

## 4. Discussion

### 4.1. Impact of Fencing and Grazing Management on the Soil Seed Bank of Different Grassland Types

Fencing management significantly promoted the accumulation of seeds in the SSB, particularly in the AD. Conversely, grazing management had a significant inhibitory effect on the seed bank, particularly on the E_seed in the AS system (Figure 5). The role of fencing may be attributed to reduced surface disturbances and soil compaction, which in turn promote the growth of parent plants and increase the likelihood of seed return to the soil [8]. This is consistent with the existing literature, that has concluded that fencing promotes the restoration of grassland seed banks [28,29]. Meanwhile, the divergent responses to fencing between the grassland types (Figure 2) may stem from the contrasting ecosystem characteristics. In the AS, higher plant productivity (AGB) and vegetation cover synergistically enhance the seed supply for the soil bank. Conversely, in the AD, limited productivity and sparse vegetation cover reduce the seed retention capacity, thereby diminishing fencing’s efficacy. This may be related to the AD’s heavier reliance on exogenous seed input under grazing, which fencing could partially restrict [15]. The higher plant productivity in the AS is a result of its relatively favorable hydrothermal conditions [30], which our experiment also corroborated (Figure 4). Higher plant productivity typically means higher vegetation cover and more species, providing more seed sources for the SSB. Higher plant productivity may also reflect the greater resource utilization efficiency (e.g., water and nutrients) of the plants in the AS ecosystem [31]. Additionally, fencing reduces surface disturbances and increases the opportunity for seeds to return to the soil. In contrast, the extreme environmental conditions in the AD, such as drought and low precipitation, limit plant productivity, resulting in lower vegetation cover. Furthermore, seed bank formation in the AD largely depends on both exogenous seed input and local vegetation protection. While fencing reduces surface disturbances and soil compaction [8], its limited capacity to fully exclude grazing pressure in the AD may reduce the native seed production (due to incomplete vegetation protection) and restrict external seed dispersal pathways [15], collectively diminishing fencing’s efficacy.

During grazing, the trampling behavior of livestock directly causes physical damage to the soil surface [32,33], leading to seeds being pressed into deeper soil layers where they are less likely to germinate, or directly impairing seed viability. At the same time, livestock’s selective grazing of certain plants may reduce the seed input of certain species [29], thereby decreasing the species diversity and evenness of the seed bank (such as the negative correlation of the E_seed in the grazing mode of the AS; Figure 5). This decline in diversity metrics is driven by the dual pressures of selective grazing—which disproportionately reduces seed inputs from palatable plant species—and trampling-induced soil compaction that physically displaces seeds or buries them below viable germination depths [8,32,33]. Moreover, grazing leads to soil compaction through trampling, reducing soil porosity and enhancing surface runoff, which decreases water infiltration, weakens the water retention capacity, and causes nutrient loss (such as the reduction in the SAP). This, in turn, further limits the formation and renewal of the seed bank.

### 4.2. The Regulatory Role of Hydrothermal Environmental Factors on Soil Seed Banks of Different Grassland Types

The contrasting hydrothermal drivers of the seed bank dynamics between the grassland types may reflect their divergent adaptive strategies (Figure 3 and Figure 4). The size of the standing seed bank (SSB) is governed by a balance between the seed input (e.g., dispersal and reproduction) and output (e.g., germination and decay). Hydrothermal and soil conditions primarily regulate the SSB by modulating the seed output through germination. Increased precipitation may directly promote seed germination and accumulation by enhancing the soil moisture content [34], while rising temperatures may lead to intensified water evaporation or increased microbial activity, thereby inhibiting seed bank formation [35,36]. In the AS system, fenced management may enhance the soil moisture retention by reducing soil compaction and evaporation, further amplifying the positive effect of precipitation on the seed bank while mitigating the negative impact of high temperatures on water loss. In contrast, the seed bank in the AD system is less sensitive to water–heat factors (Figure 4), which may be related to its lower vegetation cover and limited seed sources [37]. On the one hand, the AD system is under long-term drought stress, and the plant community’s succession and ecological adaptation characteristics are constrained by precipitation over time. The extremely low precipitation leads to a scarcity of surface water resources, making it difficult for vegetation to rely on short-term rainfall events to complete seed reproduction, seed germination, and seed bank accumulation. On the other hand, the plants in the AD system are mainly drought-tolerant species (e.g., deep-rooted plants or short-lived herbaceous plants), which may cope with drought stress by reducing seed production or delaying seed germination (prolonging seed dormancy) [38]. This results in a lower dependency of the seed bank formation on precipitation, thereby reducing the responsiveness of seed density to precipitation factors (such as the GSPre). Furthermore, the indirect regulatory effect of fencing on the water–heat factors is more limited in the AD system, likely due to the low potential for vegetation recovery and the extreme scarcity of water–heat resources.

### 4.3. The Regulatory Role of Soil Factors on Soil Seed Banks of Different Grassland Types

In the AS system, the seed density shows a significant negative correlation with the STP, SAK, and NO3_N, and a significant positive correlation with NH4_N (Figure 3). This suggests that the seed bank formation in the AS system is more easily influenced by the integrated regulation of multiple factors. Fencing management significantly enhances the positive effect of NH4_N on the seed density by reducing surface disturbances and nutrient loss [39], while alleviating the negative impacts of the STP, NO3_N, and SAK on the seed bank [36,39]. This may be because fencing reduces the consumption of competitive nutrients, such as the STP, and inhibits the loss of NO3_N, thereby optimizing plant resource allocation and promoting seed bank accumulation.

In the AD system, the response of the seed bank density to the soil factors was generally weak, with only a significant positive correlation observed for certain factors, such as the SAP and SAK (Figure 3). This suggests that readily available nutrients play a key role in seed bank formation in the resource-poor desert steppe [40]. Fencing mainly promotes seed bank accumulation by maintaining the stability of the SAP and SAK. At the same time, it may indirectly enhance water and nutrient utilization efficiency by reducing soil compaction and evaporation, thereby amplifying the positive effects of readily available nutrients on seed bank formation [8,31]. However, other factors, such as the pH and SOM, did not show significant effects, indicating that the AD seed bank is more limited by long-term environmental stresses (e.g., drought and low nutrients), with fencing’s regulatory effect being relatively weak.

Additionally, soil nutrients are also dependent on the ecosystem’s water and thermal conditions. The AS system, with relatively favorable water and thermal conditions, allows for soil nutrients to be more effectively utilized by plants. Fencing, by enhancing soil water retention, may indirectly improve the effectiveness of precipitation in the AS system, further amplifying the positive impact of environmental factors on seed bank formation. In contrast, the long-term scarcity of water and thermal resources in the AD system limits the availability of soil nutrients, weakening the impact of soil factors and making fencing’s regulatory effect less pronounced in the AD system [41]. This further leads to differences in the regulatory mechanisms of the soil factors between the two grassland types.

## 5. Conclusions

This study reveals that the soil seed bank dynamics in the Tibetan grasslands are governed by ecosystem-specific adaptation strategies shaped by long-term climatic pressures. In the AS, the seed banks exhibit pronounced sensitivity to hydrothermal fluctuations—thriving under increased precipitation but declining sharply with rising temperatures. This responsiveness stems from the AS’s dense vegetation canopy, which enhances moisture retention, creating a microclimate where seed accumulation closely tracks with seasonal rainfall patterns. Conversely, the AD demonstrates resilience to short-term climatic variability, as its seed banks remain largely decoupled from annual precipitation and temperature shifts. This insensitivity reflects evolved survival strategies: deep-rooted perennials can access stable groundwater reserves, while ephemeral plants invest in long-term seed dormancy, awaiting rare germination windows. Such divergent adaptations highlight the fundamental trade-off between ecological opportunism (AS) and stress tolerance (AD), reshaping our understanding of seed bank persistence across aridity gradients.

The management interventions yielded context-dependent outcomes. Fencing universally enhanced seed bank density but achieved maximal efficacy in the AS, where reduced soil disturbances synergized with natural moisture retention to amplify the precipitation benefits. In contrast, the AD’s nutrient-poor soils (particularly phosphorus limitation) and extreme aridity constrained the fencing’s impact, underscoring the need for targeted amendments (e.g., strategic reseeding) in severely degraded systems. Critically, the seed bank–vegetation relationships served as ecological indicators: their strong correlations in the AS signaled intact regenerative potential, whereas weak linkages in the AD revealed degradation thresholds beyond which passive recovery becomes improbable.

These findings challenge uniform restoration paradigms. For the AS, fencing alone may suffice to leverage the inherent hydrological connectivity and seed–vegetation feedbacks. For the AD, combining fencing with active seed introduction and soil enrichment is imperative to bypass the ecological bottlenecks. Future research should quantify seed longevity under aridification and identify the optimal nutrient amendments for AD recovery, enabling predictive models of grassland resilience. By aligning restoration practices with these ecosystem-specific strategies, the stakeholders could optimize the limited resources to safeguard the Tibetan Plateau’s ecological functions.

## Figures and Tables

**Figure 1 plants-14-00900-f001:**
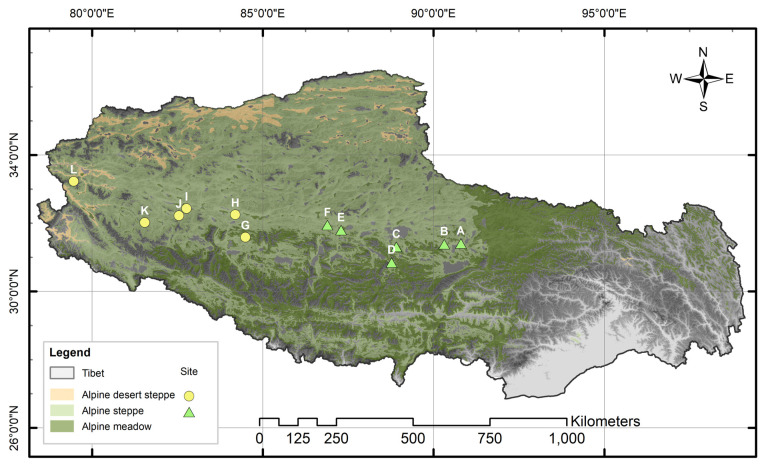
Geographical location of the study area (A–L sites).

**Figure 2 plants-14-00900-f002:**
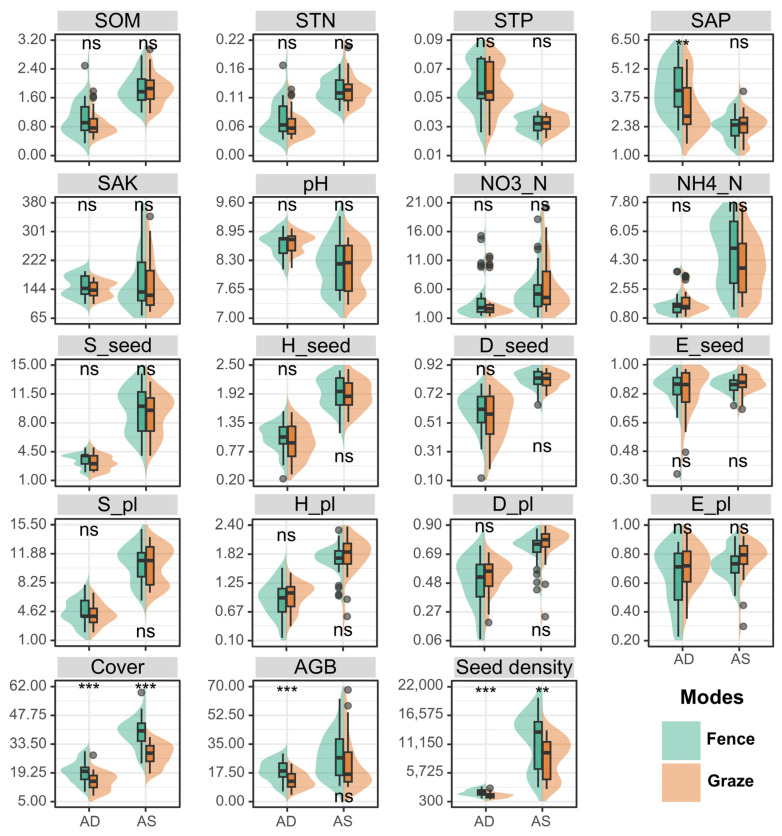
Differences in observed indicators of alpine desert steppe and alpine steppe under grazing and fenced management modes. Non-significant differences are not noted (*p* > 0.05). SOM, soil organic matter (%); STN, soil total nitrogen (g/kg); STP, soil total phosphorus (g/kg); SAP, soil available phosphorus (mg/kg); SAK, soil available potassium (mg/kg); NO3_N, nitrate nitrogen (mg/kg); NH4_N, ammonium nitrogen (mg/kg); S_seed, species richness of SSB; H_seed, Shannon–Wiener index of SSB; D_seed, Simpson index of SSB; E_seed, Pielou evenness index of SSB; S_pl, species richness of aboveground vegetation; H_pl, Shannon–Wiener index of aboveground vegetation; D_pl, Simpson index of aboveground vegetation; E_pl, Pielou evenness index of aboveground vegetation; Cover (%); AGB, aboveground biomass. The significance levels for each factor are indicated as ns *p* > 0.05, ** *p* < 0.01, and *** *p* < 0.001. Black dots indicate outliers.

**Figure 3 plants-14-00900-f003:**
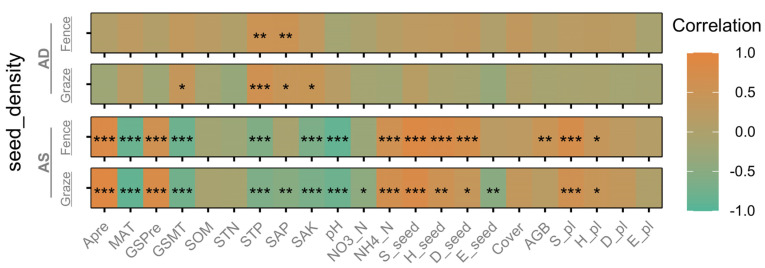
Soil seed bank density and its correlation (Spearman) with various environmental factors. Non-significant differences are not noted (*p* > 0.05). APre, annual precipitation (mm); MAT, mean annual temperature (°C); GSPre, growing season precipitation (May to September, mm); GSMT, growing season mean temperature (May to September, °C); SOM, soil organic matter (%); STN, soil total nitrogen (g/kg); STP, soil total phosphorus (g/kg); SAP, soil available phosphorus (mg/kg); SAK, soil available potassium (mg/kg); NO3_N, nitrate nitrogen (mg/kg); NH4_N, ammonium nitrogen (mg/kg); S_seed, species richness of SSB; H_seed, Shannon–Wiener index of SSB; D_seed, Simpson index of SSB; E_seed, Pielou evenness index of SSB; Cover (%); AGB, aboveground biomass; S_pl, species richness of aboveground vegetation; H_pl, Shannon–Wiener index of aboveground vegetation; D_pl, Simpson index of aboveground vegetation; E_pl, Pielou evenness index of aboveground vegetation. The significance levels for each factor are indicated as * *p* < 0.05, ** *p* < 0.01, and *** *p* < 0.001.

**Figure 4 plants-14-00900-f004:**
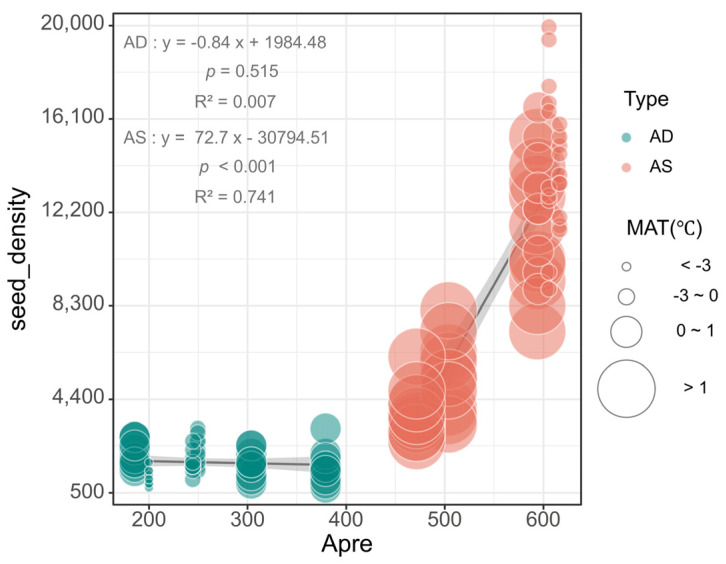
The response of the seed density to hydrothermal factors in the two grassland types. The sizes of the bubbles in the figure represent the MAT values corresponding to the different seed densities in the AS and AD, indicated by two distinct colors. APre, annual precipitation (mm); MAT, mean annual temperature (°C).

**Figure 5 plants-14-00900-f005:**
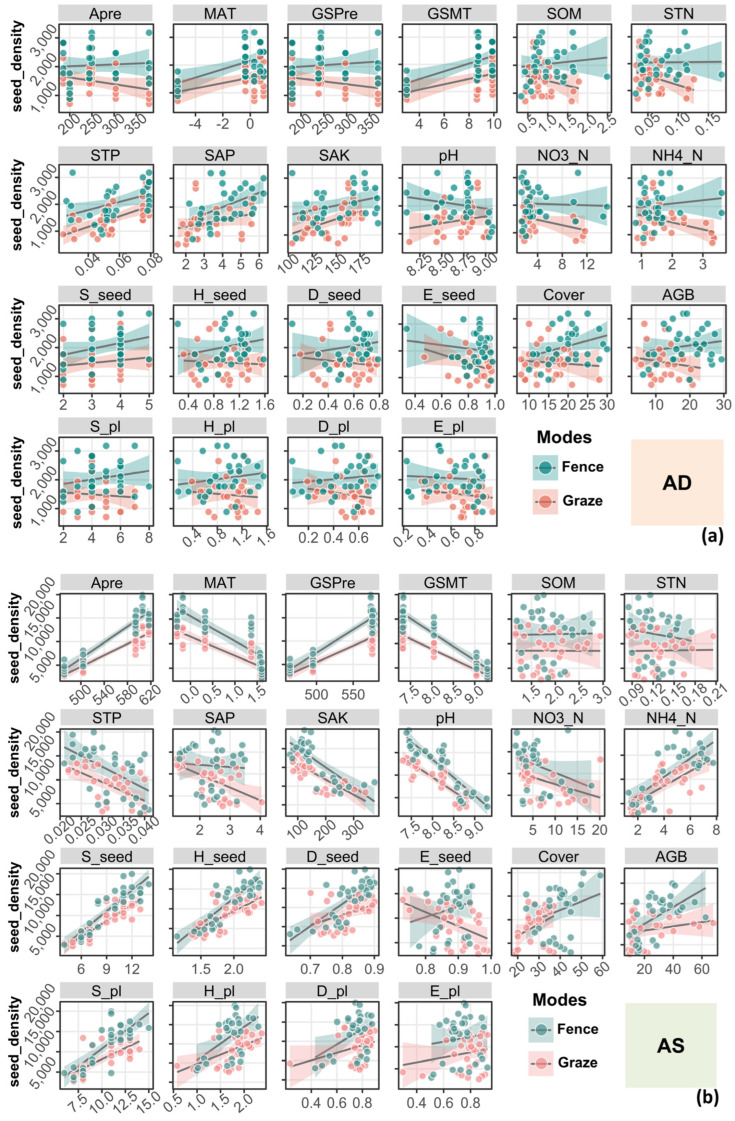
Seed density relationships and influencing factors under grazing and fenced management modes. (**a**) In alpine desert steppe (AD); (**b**) in alpine steppe (AS). APre, annual precipitation (mm); MAT, mean annual temperature (°C); GSPre, growing season precipitation (May to September, mm); GSMT, growing season mean temperature (May to September, °C); SOM, soil organic matter (%); STN, soil total nitrogen (g/kg); STP, soil total phosphorus (g/kg); SAP, soil available phosphorus (mg/kg); SAK, soil available potassium (mg/kg); NO3_N, nitrate nitrogen (mg/kg); NH4_N, ammonium nitrogen (mg/kg); S_seed, species richness of SSB; H_seed, Shannon–Wiener index of SSB; D_seed, Simpson index of SSB; E_seed, Pielou evenness index of SSB; Cover (%); AGB, aboveground biomass; S_pl, species richness of aboveground vegetation; H_pl, Shannon–Wiener index of aboveground vegetation; D_pl, Simpson index of aboveground vegetation; E_pl, Pielou evenness index of aboveground vegetation. Regression fitting results are presented in Table 2.

**Table 1 plants-14-00900-t001:** Basic overview of survey sites along alpine grassland transect across northern Tibet.

Site	Type	Longitude	Latitude	Altitude	APre	MAT	GSPre	GSMT
A	AS	90.80	31.41	4650	616.63	−0.30	579.06	7.35
B	AS	90.31	31.39	4650	605.74	−0.17	574.01	7.34
C	AS	88.91	31.31	4720	595.06	0.33	576.84	7.32
D	AS	88.77	30.85	4700	593.92	1.37	576.01	8.05
E	AS	87.29	31.80	4580	503.85	1.57	495.30	9.04
F	AS	86.89	31.95	4500	471.45	1.60	463.90	9.30
G	AD	84.49	31.59	4600	379.15	0.30	369.16	8.64
H	AD	84.19	32.25	4500	303.81	0.98	297.54	9.90
I	AD	82.76	32.43	4480	249.56	−0.38	243.52	8.74
J	AD	82.54	32.22	4470	244.49	−0.34	238.54	8.76
K	AD	81.54	32.02	4600	185.41	0.74	180.71	9.87
L	AD	79.46	33.23	4330	199.97	−5.29	180.85	2.98

Note: altitude (m); APre, annual precipitation (mm); MAT, mean annual temperature (°C); GSPre, growing season precipitation (May to September, mm); GSMT, growing season mean temperature (May to September, °C); AS, alpine steppe; AD, alpine desert steppe.

**Table 2 plants-14-00900-t002:** General linear regression results of seed density and influencing factors under grazing and fenced management modes. APre, annual precipitation (mm); MAT, mean annual temperature (°C); GSPre, growing season precipitation (May to September, mm); GSMT, growing season mean temperature (May to September, °C); SOM, soil organic matter (%); STN, soil total nitrogen (g/kg); STP, soil total phosphorus (g/kg); SAP, soil available phosphorus (mg/kg); SAK, soil available potassium (mg/kg); NO3_N, nitrate nitrogen (mg/kg); NH4_N, ammonium nitrogen (mg/kg); S_seed, species richness of SSB; H_seed, Shannon–Wiener index of SSB; D_seed, Simpson index of SSB; E_seed, Pielou evenness index of SSB; Cover (%); AGB, aboveground biomass; S_pl, species richness of aboveground vegetation; H_pl, Shannon–Wiener index of aboveground vegetation; D_pl, Simpson index of aboveground vegetation; E_pl, Pielou evenness index of aboveground vegetation. The significance levels for each factor are indicated as * *p* < 0.05, ** *p* < 0.01, and *** *p* < 0.001.

Observed Variables	Modes	Alpine Desert Steppe					Alpine Steppe				
		R^2^	*p*		Slope	Intercept	R^2^	*p*		Slope	Intercept
APre	Fence	0.005	*p* > 0.05		0.63	1887.72	0.873	*p* < 0.001	***	84.56	−35,832.26
	Graze	0.081	*p* > 0.05		−2.30	2081.25	0.861	*p* < 0.001	***	60.84	−25,756.75
MAT	Fence	0.282	*p* < 0.01	**	151.75	2153.03	0.661	*p* < 0.001	***	−5067.00	15,614.41
	Graze	0.148	*p* < 0.05	*	95.53	1545.29	0.731	*p* < 0.001	***	−3861.97	11,419.49
GSPre	Fence	0.012	*p* > 0.05		1.03	1794.09	0.848	*p* < 0.001	***	99.48	−42,239.40
	Graze	0.058	*p* > 0.05		−1.93	1966.64	0.815	*p* < 0.001	***	70.65	−29,862.71
GSMT	Fence	0.309	*p* < 0.01	**	142.82	888.27	0.870	*p* < 0.001	***	−5707.69	57,937.27
	Graze	0.195	*p* < 0.05	*	98.62	678.12	0.870	*p* < 0.001	***	−4135.05	41,940.83
SOM	Fence	0.014	*p* > 0.05		155.32	1891.87	0	*p* > 0.05		164.63	11,592.13
	Graze	0.051	*p* > 0.05		−348.49	1784.74	0	*p* > 0.05		−0.08	8584.95
STN	Fence	0	*p* > 0.05		210.67	2037.21	0.015	*p* > 0.05		−25,930.49	15,105.43
	Graze	0.113	*p* > 0.05		−6993.11	1899.02	0	*p* > 0.05		1902.19	8346.84
STP	Fence	0.176	*p* < 0.05	*	15,210.17	1179.56	0.286	*p* < 0.01	**	−458,503.30	26,490.93
	Graze	0.369	*p* < 0.001	***	19,599.59	380.92	0.368	*p* < 0.001	***	−392,218.90	21,279.62
SAP	Fence	0.223	*p* < 0.01	**	249.24	1001.58	0.004	*p* > 0.05		−503.49	13,100.26
	Graze	0.085	*p* > 0.05		123.41	1071.87	0.280	*p* < 0.01	**	−2947.60	15,769.09
SAK	Fence	0.098	*p* > 0.05		6.69	1032.69	0.607	*p* < 0.001	***	−52.07	20,409.00
	Graze	0.193	*p* < 0.05	*	11.04	−95.73	0.593	*p* < 0.001	***	−40.12	14,763.02
pH	Fence	0.040	*p* > 0.05		−456.35	6012.49	0.800	*p* < 0.001	***	−8018.45	77,646.56
	Graze	0.038	*p* > 0.05		487.35	−2764.01	0.727	*p* < 0.001	***	−6272.07	59,718.90
NO3_N	Fence	0.002	*p* > 0.05		−7.10	2082.96	0.098	*p* > 0.05		−410.02	14,283.61
	Graze	0.085	*p* > 0.05		−48.77	1666.41	0.152	*p* < 0.05	*	−308.96	10,684.39
NH4_N	Fence	0.015	*p* > 0.05		117.59	1866.78	0.559	*p* < 0.001	***	1931.11	2761.24
	Graze	0.124	*p* > 0.05		−248.51	1930.78	0.508	*p* < 0.001	***	1527.63	2526.54
S_seed	Fence	0.082	*p* > 0.05		206.53	1336.15	0.788	*p* < 0.001	***	1651.79	−3851.86
	Graze	0.026	*p* > 0.05		94.89	1181.27	0.765	*p* < 0.001	***	1291.86	−2912.77
H_seed	Fence	0.045	*p* > 0.05		434.82	1598.76	0.701	*p* < 0.001	***	13,184.72	−13,676.36
	Graze	0.004	*p* > 0.05		−100.45	1577.67	0.429	*p* < 0.001	***	8383.63	−7385.63
D_seed	Fence	0.027	*p* > 0.05		709.59	1636.04	0.585	*p* < 0.001	***	58,364.01	−35,751.76
	Graze	0.021	*p* > 0.05		−476.21	1742.21	0.158	*p* < 0.05	*	26,631.38	−13,216.23
E_seed	Fence	0.017	*p* > 0.05		−652.00	2607.14	0.042	*p* > 0.05		21,984.11	−7363.92
	Graze	0.144	*p* < 0.05	*	−1565.07	2802.83	0.234	*p* < 0.01	**	−28,695.10	34,182.31
Cover	Fence	0.099	*p* > 0.05		36.71	1322.83	0.080	*p* > 0.05		183.51	4707.97
	Graze	0.007	*p* > 0.05		−9.43	1624.10	0.136	*p* < 0.05	*	243.42	1744.61
AGB	Fence	0.026	*p* > 0.05		15.78	1760.15	0.291	*p* < 0.01	**	204.52	6021.44
	Graze	0.029	*p* > 0.05		−18.59	1721.46	0.050	*p* > 0.05		52.02	7376.79
S_pl	Fence	0.039	*p* > 0.05		75.44	1707.59	0.642	*p* < 0.001	***	1751.91	−6733.35
	Graze	0.008	*p* > 0.05		−34.44	1629.83	0.499	*p* < 0.001	***	1111.95	−2942.44
H_pl	Fence	0.029	*p* > 0.05		291.61	1786.56	0.275	*p* < 0.01	**	8683.84	−2957.55
	Graze	0.008	*p* > 0.05		−178.42	1656.72	0.260	*p* < 0.01	**	4938.36	−245.76
D_pl	Fence	0.015	*p* > 0.05		391.44	1864.83	0.182	*p* < 0.05	*	19,997.50	−2812.51
	Graze	0.023	*p* > 0.05		−632.04	1813.43	0.125	*p* > 0.05		9829.71	1117.23
E_pl	Fence	0.005	*p* > 0.05		−206.58	2182.30	0.013	*p* > 0.05		5470.33	7902.14
	Graze	0.015	*p* > 0.05		−443.85	1795.95	0.042	*p* > 0.05		5707.17	4193.77

## Data Availability

The data will be made available on request.

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
