# Peer review of "Fencing vs. Grazing: Divergent Effects on Soil Seed Bank Structure and Grassland Recovery Pathways in Northern Tibetan Alpine Grasslands"

_plants, 2025, doi:10.3390/plants14060900_

Round 1
Reviewer 1 Report
Comments and Suggestions for Authors
The manuscript presents valuable insights and offers meaningful contributions to the study of mountain grassland dynamics; however, there are some points that could be improved to enhance the clarity and robustness of the work. Specifically, I suggest providing a clearer and more detailed description of the ecological and vegetation characteristics of the two site groups analysed in this study, particularly differentiating the two types of grasslands (AS vs. AD habitats).
The discussion on the lack of significant responses to vegetation biomass and climate in the desert steppe (AD habitat) seems a little weak. I recommend strengthening this section by exploring the implications of soil seed bank sensitivity in the AD system in more details. Although the Authors adequately discuss the challenges that vegetation faces in this climate complicating the analysis of its response to treatment and suggest that soil nutrients economy rather than mean precipitation or temperature may explain the lack of management influences, a deeper exploration of the ecological and dynamic differences between the two communities would be beneficial to enrich the discussion.
Additionally, in the results section, an approach that considers the distribution of responses along the transect and show how these vary if any could provide more insight, given the vast and heterogeneous nature of the mountain area considered in the study. Reducing all responses to a single value may overlook local variations? This point if not amendable should at least be briefly addressed in discussion. Overall, the study holds good potential, but would benefit from minor revisions to clarify these aspects.
Main questions:
Methods
Line 99 Please provide more context on the aspect and dominant vegetation by acknowledging the differences between the two ecosystems and hints at the vegetation in each in the sentence “major AS and alpine desert steppe (AD)”
Line 126: fenced grazing means that grazing was avoided?
Lines 128 and 135: how many vegetation survey were conducted in each plot of 10x10m? is that one per plot? And in total?
Please see lines: “Within each plot, the SSB, vegetation, soil, and other factors were 128 surveyed and sampled. “ and “In the eastern meadow sites of Northern Tibet, the sampling area was 0.5×0.5 m, while in the central steppe and western desert regions, the sampling area was 1×1 m.
Lines 147-151 For the following equations on plant diversity employed by the Authors, main references are missing, please provide Species richness ( ), Shannon-Wiener index ( , Eq.2), Simpson index ( , Eq.3) 147 and Pielou evenness index ( , Eq.4) S H DE
Line 153: Please give details on how many soil samples were taken within each plot or subplot; what is community here? Please see words: “within each community[…]”
Line 160-170. Here the Authors should write at which condition seed germination was conducted?
Lien 160: according to me here the sentence is: We estimated the species composition of SSB by scoring germinated seeds by the following method”…. And please avoid ” based on germinated indivdiuals” when it is obvious […] for example write: […] using the Flora of Tibet we were able to identify..
Line 166. The following sentence is unprecise -not clear, it must be rewritten and the Authors should say clearly how many soil samples were scored, the total of seedlings per species etc : The calculation methods for SSB indicators are consistent with those for plant community characteristics.
Other minor imprecisions are SSB density and various factors“ of lines 200-201? And on line 203 SSB density and seed density?
Results
Line 221 Fig 1 seed density; the sentence is not very clear to me and specifically the Author should discuss the impressive difference between the seed amounts in the two habitats “the seed density of AS was higher than that 221 of AD, indicating that the seed bank of AS is more sensitive to management modes.
Discussion
Starting from discussion, the Authors should as much as possible abandon the use of acronyms in their variables; for example: Line 329 for example E-seed could stand for “seed diversity eveness”? please correct this and lines 429 In the AS system, seed density showed a significant positive correlation with APre 429 and GSPre, and a significant negative correlation with MAT and GSMT. In contrast, in the AD system,[…]
Line 405 the sentence: whereas the impact of fencing is relatively smaller in AD due to its lower vegetation cover Because, while it is true that the greater plant biomass, the most seeds enter the soil seed bank, and that this response is highly sensitive to grazing, it is not clear why, if the biomass decreases, the soil seed bank that forms should not be also affected by livestock grazing; now I know that this is discussed later in the paper however, a brief comment could be anticipated here.
Line 416: A second sentence to be rethought a bit is the following: "Furthermore, seed bank formation in AD largely depends on the input of exogenous seeds, and the protective role of fencing may actually restrict this external seed input [15]." In my view, the effect of fencing could be reduced also because it does not protect the vegetation from grazing, which is where the seeds originate from.
Line 421: such as the 421 negative correlation of E_seed in the grazing mode of AS; Figure 5). Line 421: This argumentation seems less convincing to me if only the E_seed variable is affected. In fact, what can we conclude about the remaining parameters of seed diversity? This aspect is either scarcely addressed or not discussed at all, which I believe is a weakness of the paper. However, it does not necessarily diminish the value of these findings.
Conclusion
In this section also, please replace variable acronyms with full names. see for example line 487
Comment: As already indicated inmy general comment, It would be important to discuss more thoroughly here only briefly and more extensively in Discussion, the factors influencing results obtained in the desertic steppe (AD), specifically regarding the low dependency of soil seed bank density and variability of the seeds of this habitat on vegetation cover and climatic parameters. A deeper exploration of the floristic and vegetation composition of this ecosystem could help explain why the seed bank in the AD system shows a weaker response to vegetation and water-thermal conditions compared to the AS system. For example, can we have some hints from in field vegetation surveys run by the Authors (maybe in Supplemental materials)? Understanding if specific plants and their site adaptations, also in a broader ecological context of the AD habitat could offer insights into the relatively limited effect of climate and vegetation on seed bank dynamics in this region.
Reviewer 2 Report
Comments and Suggestions for Authors
General opinion:
In the publication, the authors conducted a comparative study of the vegetation and seed bank of the northern Tibetan alpine grasslands. They examined the effect of grazing exclusion as a function of environmental factors. The study contains important research results for alpine grassland management and nature conservation. A well-designed and implemented study with good results. However, the article still needs corrections before it can be published.
The introduction and the material and methods are somewhat incomplete, and the parts of the results and discussion chapters are mixed up in places. The illustrations are of good quality, but from a didactic point of view, most of them need improvement. Throughout the article, the order of the figures should be consistent with that in the text. I suggested this based on Figure 2 (e.g. AD then AS) and in the rest of the article. This can of course be in reverse order (AS then AD), but then it should be consistent in all figures and in the text. The discussion needs to be thoroughly reviewed, conclusions need to be drawn, not just a summary.
Critical comments:
The title needs to be corrected as it does not refer to the study that was conducted. There is no mention of a pattern here.
Abstract:
Needs improvement because the methods section is incomplete and the results are not well highlighted.
Some specifics:
The "pattern" in line 20 is not good, instead of human impacts, grazing is more specific. Not 12 pairs, but six pairs, i.e. 12 sites, were examined.
Line 21: I think the effect of excluding grazing was studied in two types of grassland.
Line 24: There was only one type of treatment.
Line 27: (E_seed)- the abbreviation is not explained, it would be better if it was written out.
Line 35: Replace the keywords that are already in the title with new ones.
Line 99: Either write out both vegetation types, or just the abbreviations, but mixing them is not good.
Figure 1: The figure is very nice, but the location of the sites within the vegetation unit is not clearly visible. I propose to publish a small presentation of the outline of the Tibet image without coloring. On the map showing vegetation types, zoom in on the area covered by the sites. Make the colors more contrasting. For example, now it looks like site D is in the "alpine meadow" area, which is incorrect.
Table 1: It is not clear what time interval the climate data refers to. Given climate change, this needs to be communicated. Why is AM included in the explanation when it is not in the table?
The L site is quite different from the other areas, both in terms of distance and climatic data.
In section 2.1. test sites, the two vegetation types should be presented in comparison. The macroclimatic difference can already be guessed from the table provided, but it is blurred in the current description. There is a significant climate gradient along the transect, which may also cause a gradual change in vegetation type. This is what will be presented in this chapter. A schematic presentation of vegetation and soils was completely omitted. At least the dominant species and soil types should be given.
Line 120. Complete the first table with a vegetation column (e.g. association name or at least dominant species) and a soil type column.
Chapter 2.3.: How was the visual vegetation cover estimate made? Was the % cover estimated?
Line 134: What kind of alpine meadow sampling is this?
Chapter 2.4: Please clarify the location of the soil sampling method, it is currently unclear. Were samples taken from all sites?
Chapter 2.5: Distilled water was used for aqueous pH measurement, right?
Results chapter:
Figure 2 is too complicated and not didactic. I suggest a complete re-arrangement of the figure as follows:
SOIL: SOM, pH, STP, SAP,
STN, NO3_N, NH4_N, SAK
SEED: Seed density, H_seed, D_seed,E_seed,
PLANTt: Cover; H_pl, D_ pl, E_ pl,
AGB, Modes (Fence, Gaze )
Do you need the S_seed, S_pl data? This is included in the diversity indices. These are species numbers, as correctly stated in line 150, and not species richness or diversity. These need to be corrected in the figure explanations and in the text (also in line 147).
The order in the figure is AD, AS, so the order should be reversed in the legend. It is not clear how many sampling pairs were analyzed.
Line 245: the period after "noted" should be deleted.
The results should follow the order of the re-arranged figure, omitting the explanations, which are included in lines 225, 229, for example. These should not be moved here, but to the discussion.
At the beginning of line 254: As shown in Figure 3,
Line 256: Correct order AD then AS. This also applies to the rest of the results.
Figure 3: I recommend keeping the order suggested in Figure 2, and also swapping the order of "MAT" and "GSPpre" in the figure.
The order of the lawns should also be consistent, so AD is on the top and AS is on the bottom.
Line 266: Spearman should be placed after the word correlation.
Line 267: the period after "noted" should be deleted.
The annual interval for calculating climatic data must be supplemented.
The presentation of the results in Figure 3 should be corrected. It contains explanations, e.g. 258-259,296, 306-307, 315-316, and incorrect statement in line 260.
Figure 4: 341. row. Apre (mm)
I suggest plotting the thousand seed density values, there are too many zeros.
Line 338: "Increase" was omitted from the temperature.
Table 2: For the order of observed variables, see the suggestions in Figure 3.
Figure 5: For the order of observed variables, see the suggestions in Figure 3. I also recommend plotting the thousand seed density values here. Furthermore, it would be useful to place the AD and AS figures one below the other, because they are evaluated in comparison.
Chapter 3.3, line 373: The correct order is AD and AS systems.
Lines 373-377: The results should also be AD and then AS.
Line 379: Should be moved to discussion.
Chapter 3.4: The order here should be AD then AS, as before.
4. Discussion:
There is little comparison with literature.
396-397.: Density cannot accumulate (accumulation of SSB density). Also, it's too complicated.
My suggestion: Fencing management significantly promoted the accumulation of seeds in the soil, exspecially in the alpin desert steppe.
Lines 397-399: It should be worded more simply.
Lines 399-402: It should be compared to literature, condensed, and paraphrased.
Lines 404-405: plant productivity or vegetation cover?? Need clarification.
Unfortunately, this chapter also has an AS and AD order.
414-415.: Arguable, because the proportion of anemochorous seeds may be large, which is hardly limited by this enclosure.
417-426.: The effect of grazing is more complex. The grazing animal also plants by trampling it. The effect may depend strongly on the extent of grazing. Abandoning grazing may even lead to a decrease in diversity. This can only be assessed by examining grazing levels.
429-432.: This is a result, not a discussion.
433-434.: Seed banks and seed germination should not be confused! This is not an explanation.
439-440.: I don't understand this explanation. The hydrothermal tolerance of AD may be greater.
441-453.: It should be supported by the literature that the seed dormancy of AD species (or at least their dominant species) is greater than that of AS species, and that they are more deeply rooted. It is also questionable to what extent herbivores that are not excluded from grazing on the two types of grassland have different access to and consumption of the seeds. It is necessary to incorporate relevant literature into this.
446-447: (e.g.,deep-rooted plants or short-lived herbaceous plants). What do you mean by this? Too imprecise.
461-465: Too speculative, and also unproven, because no pH decrease was shown in the fenced areas.
The conclusion chapter needs to be revised because in its current form it is a summary.
My questions:
1. It was not stated whether the enclosure only excludes large animals, or e.g. small mammals (e.g. marmots) as well? Is it possible that their role is different in fenced/grazed and AD/AS areas?
2. Why is there no difference in biomass between fenced and grazed areas if there is cover?
3. Is the time elapsed since the enclosure sufficient for the soil parameters of the examined vegetation units to change significantly (see e.g. Figure 2)? Why would the rate of change be the same in the two types of grassland?
4. The study sites appear to be located along a climate gradient. How was this taken into account?
5. Nothing is written about dominant plant species, although the ratio of annual/perennial grass species is important from the point of view of the soil seed bank. The depth of the root system is also important, especially in the case of dominant species. It can be assumed that the role of the seed bank is also of different importance in the two types of grasslands. In the case of perennial species, vegetative reproduction and clonal nature may be more important for regeneration than the soil seed bank. Also, what do you know about the proportion of seeds that fall?
Reviewer 3 Report
Comments and Suggestions for Authors
Manuscript plants-3447572-peer-review-v1
The submitted manuscript presents research focused on soil seed bank patterns and the influence of environmental conditions, including fencing and grazing management. A scientific paper focusing on soil seed bank patterns should clearly justify the significance of the research problem in the introduction. Starting with a characterization of the study area is illogical. The authors support some of their claims with irrelevant references, for example:
- Lines 42-43: The decline of ecological functions is based on citation #5, which addresses ecophysiological response of wetlands
- Line 49: Buffering against ecological disturbances is based on citation #9, which deals with weed ecology in soil
Line 51: Why do the authors believe that "the cold conditions prolong seed longevity and reduce seed decomposition rates" is a crucial entry point for understanding and restoring the dynamics of alpine grassland ecosystems? This conclusion is not explicitly supported by citation #10.
Lines 58-59: There are many scientific studies, including books, devoted to understanding the relationships between environmental factors (e.g., climate, soil nutrients) and SSB dynamics, such as:
- Leck M.A., Parker V.T. & Simpson R.L. (1989): Ecology of soil seedbanks. Academic Press, San Diego
- Fenner M. (1995): Ecology of Seed Banks. Routledge
Lines 61-65: Please provide additional citations and detailed results from studies addressing SSB and soil nutrients influence, and other environmental factors including regulation of the density and diversity of the SSB (e.g., citations 11-17)
Lines 67-73 and 77-80 are superfluous.
The introductory chapter is based on a limited number of citations or irrelevant sources. The introduction lacks consistency and fails to clearly justify the significance of the research problem – why the topic is important and why these two specific questions need to be addressed. This work is not the first to address SSB in relation to hydrothermal and soil conditions. The introduction needs to be revised accordingly and supplemented with relevant sources on the studied topic.
Methods
Line 100: What were the criteria for selecting the transect and 12 research plots? Are environmental conditions (e.g., soil, vegetation, habitat) homogeneous? Was the goal to select different or similar plots in terms of environmental conditions?
Lines 134-135: Why were two plot sizes chosen (0.5x0.5 and 1x1 m)? How was this difference addressed in the analysis?
Line 136: How was the visual assessment conducted?
Line 142: How was relative cover (C) evaluated in the laboratory?
Lines 147-149: Why did you use 4 diversity formulas (Species richness, Shannon-Wiener index, Simpson index, and Pielou evenness index)? How were they used in the analysis and interpretation?
Line 158: How was germination identification and counting of species in the SSB performed? Please provide a detailed description of conditions and procedures. Is it possible that some seeds present in the soil did not germinate?
Lines 160-161: Please describe in detail and cite the germination identification method.
Line 162: Please cite the Flora of Tibet.
Line 166: Please describe in detail the calculation methods for SSB.
Lines 198-213: Please clarify the testing methods for differences between SSB and observed environmental factors.
A large amount of field and laboratory data was collected and analyzed but not fully utilized in the paper. The grazing management and its intensity are not clearly defined. How long had the plots been under fencing? The methodology is described too briefly and inadequately. The experiment cannot be replicated and evaluated based on this description. Please provide additional details.
Results
The results lack references to statistical tests for individual environmental indicators and SSB AS and AD, Table 2.
Personal assumptions, explanations, and recommendations should be described in the discussion (e.g., Lines 224, 235-236, 296).
Line 228: Which soil factors differed between the two grassland types?
Lines 254-255: Please show the results that indicate significant differences in the influence of hydrothermal factors, soil factors, plant-related factors, and seed-related factors on the SSB.
References to Fig. 2, Fig. 3, Fig. 4, and Fig. 5 are missing in the chapter text.
The chapter not only presents results but also comments on them. The text is unnecessarily long, unclear, and only limitedly presents the results of main findings and specific numerical values of results or statistical tests. The chapter should focus solely on specific outputs of statistical analyses that are further discussed. Comments and explanations of observed phenomena belong in the discussion chapter. The results text needs to be shortened and made more precise.
Discussion
Lines 402-406: Please rewrite in a more comprehensible manner.
Line 420: Did you prove livestock's selective grazing and the reduction of seed input for certain species? Results are missing.
Line 423: Evidence for soil compaction through trampling and nutrient loss is missing.
The discussion often repeats results and doesn't provide new explanations and deductions based on your own results and cited sources. It is poor in cited sources. Moreover, it comments on topics not covered in the manuscript, e.g., environmental conditions and ecosystem.
The chapter lacks answers to the objectives and hypotheses stated in the introduction.
The discussion can be shortened by removing text that repeats results. Conversely, it needs additional commentary on results in relation to other works on this topic. Citations need to be added, and the focus should remain on the research topic and results.
Conclusion
Lines 500-501: Seed bank density was significantly positively correlated with NH4_N and negatively correlated with STP, SAK, pH, and NO3_N. But what is the correlation with species richness?
The conclusion should be clear and concise. It should contain the main research findings and new insights for current knowledge. Management recommendations and predictions should also appear here.
Overall assessment
The paper is written inconsistently, with chapter contents not flowing together. The authors managed to collect a large amount of data, but their analysis is not entirely clear. They seem uncertain how to handle the large number of results, and the discussion fails to relate the results to the current state of knowledge. The text is too long considering how little new information it provides. The paper should contain answers to the stated hypotheses and clearly formulated new findings and contributions to current knowledge. I recommend revising the chapters according to the above recommendations and shortening the text.
Round 2
Reviewer 1 Report
Comments and Suggestions for Authors
The Authors have revised the manuscript according to the reviewers' suggestions. In particular, regarding my comments, they have responded coherently and attentively to all the observations and concerns I raised while reading the text. The discussion appears to be significantly improved. Therefore, I believe the manuscript, pending a final linguistic revision in English, is ready for publication.
Thank you
Author Response
Thank you for your dedicated efforts in reviewing our manuscript entitled "Fencing vs. Grazing: Contrasting Effects on Soil Seed Bank Structure and Environmental Drivers in Northern Tibetan Alpine Grasslands" (ID: plants-3447572). We sincerely appreciate your constructive feedback and valuable suggestions throughout the peer-review process.
Reviewer 2 Report
Comments and Suggestions for Authors
General opinion:
This article still needs some corrections before it can be published.
The serial numbers are the serial numbers of the first version of the article.
The introduction is somewhat incomplete, and the parts of the results and discussion chapters are mixed up in places. The illustrations are of good quality, but from a didactic point of view, most of them need improvement. Throughout the article, the order of the figures should be consistent with that in the text. I suggested this based on Figure 2 (e.g. AD then AS) and in the rest of the article. This can of course be in reverse order (AS then AD), but then it should be consistent in all figures and in the text.
The introduction was incomplete, and it is even more so now that it has been shortened.
Critical comments:
Line 27: (E_seed)- the abbreviation is not explained, it would be better if it was written out.
Line 99: Either write out both vegetation types, or just the abbreviations, but mixing them is not good.
Figure 1: The figure is very nice, but the location of the sites within the vegetation unit is not clearly visible. I propose to publish a small presentation of the outline of the Tibet image without coloring. On the map showing vegetation types, zoom in on the area covered by the sites. Make the colors more contrasting.
Table 1: It is not clear what time interval the climate data refers to. Given climate change, this needs to be communicated. Why is AM included in the explanation when it is not in the table?
In section 2.1.: There is a significant climate gradient along the transect, which may also cause a gradual change in vegetation type. This is what will be presented in this chapter.
Chapter 2.3.: How was the visual vegetation cover estimate made? Was the % cover estimated?
Results chapter:
Figure 2 is too complicated and not didactic. I suggest a complete re-arrangement of the figure as follows:
SOIL: SOM, pH, STP, SAP,
STN, NO3_N, NH4_N, SAK
SEED: Seed density, H_seed, D_seed,E_seed,
PLANTt: Cover; H_pl, D_ pl, E_ pl,
AGB, Modes (Fence, Gaze )
Do you need the S_seed, S_pl data? This is included in the diversity indices. These are species numbers, as correctly stated in line 150, and not species richness or diversity. These need to be corrected in the figure explanations and in the text (also in line 147).
The order in the figure is AD, AS, so the order should be reversed in the legend. It is not clear how many sampling pairs were analyzed.
The results should follow the order of the re-arranged figure, omitting the explanations, which are included in lines 225, 229, for example. These should not be moved here, but to the discussion.
Line 256: Correct order AD then AS. This also applies to the rest of the results.
Figure 3: I recommend keeping the order suggested in Figure 2, and also swapping the order of "MAT" and "GSPpre" in the figure.
The order of the lawns should also be consistent, so AD is on the top and AS is on the bottom.
Line 266: Spearman should be placed after the word correlation.
The annual interval for calculating climatic data must be supplemented.
The presentation of the results in Figure 3 should be corrected. It contains explanations, e.g. 258-259,296, 306-307, 315-316, and incorrect statement in line 260.
Figure 4: 341. row. Apre (mm)
I suggest plotting the thousand seed density values, there are too many zeros.
Table 2: For the order of observed variables, see the suggestions in Figure 3.
Figure 5: For the order of observed variables, see the suggestions in Figure 3. I also recommend plotting the thousand seed density values here. Furthermore, it would be useful to place the AD and AS figures one below the other, because they are evaluated in comparison.
Chapter 3.3, line 373: The correct order is AD and AS systems.
Lines 373-377: The results should also be AD and then AS.
Line 379: Should be moved to discussion.
Chapter 3.4: The order here should be AD then AS, as before.
4. Discussion:
There is little comparison with literature.
Lines 397-399: It should be worded more simply.
Unfortunately, this chapter also has an AS and AD order.
433-434.: Seed banks and seed germination should not be confused! This is not an explanation.
439-440.: I don't understand this explanation. The hydrothermal tolerance of AD may be greater.
441-453.: It should be supported by the literature that the seed dormancy of AD species (or at least their dominant species) is greater than that of AS species, and that they are more deeply rooted. It is also questionable to what extent herbivores that are not excluded from grazing on the two types of grassland have different access to and consumption of the seeds. It is necessary to incorporate relevant literature into this.
446-447: (e.g.,deep-rooted plants or short-lived herbaceous plants). What do you mean by this? Too imprecise.
461-465: Too speculative, and also unproven, because no pH decrease was shown in the fenced areas.
My questions:
1. It was not stated whether the enclosure only excludes large animals, or e.g. small mammals (e.g. marmots) as well? Is it possible that their role is different in fenced/grazed and AD/AS areas?
2. Why is there no difference in biomass between fenced and grazed areas if there is cover?
3. Is the time elapsed since the enclosure sufficient for the soil parameters of the examined vegetation units to change significantly (see e.g. Figure 2)? Why would the rate of change be the same in the two types of grassland?
4. The study sites appear to be located along a climate gradient. How was this taken into account?
Round 3
Reviewer 2 Report
Comments and Suggestions for Authors
The authors have made most of the corrections I suggested, and have provided acceptable answers to the remaining ones. You must have misunderstood one of my suggestions. Climatic data are average values, therefore the specified time interval is required (e.g. 1988-2018). It is not enough to include a reference (2018). I recommend the authors' article for publication.
Author Response
Thank you sincerely for your final review and constructive guidance on our manuscript "Fencing vs. Grazing: Contrasting Effects on Soil Seed Bank Structure and Environmental Drivers in Northern Tibetan Alpine Grasslands" (ID: 3447572).
Regarding the climatic data specification:
We confirm that all analyses were based on monthly meteorological records from the year 2018 (January-December), rather than multi-year averages.
This has been explicitly stated in:
line:187
"For 2018 monthly climatic records (January-December), we obtained climate information from..."
The reference to "2018" in the original submission specifically denoted the dataset's temporal scope. We have removed any ambiguous phrasing to prevent misinterpretation.
We deeply appreciate your rigorous review that enhanced the precision of this work.